# Texture Analysis in Diagnosing Skin Pigmented Lesions in Normal and Polarized Light—A Preliminary Report

**DOI:** 10.3390/jcm11092505

**Published:** 2022-04-29

**Authors:** Paweł Popecki, Kamil Jurczyszyn, Marcin Ziętek, Marcin Kozakiewicz

**Affiliations:** 1Department of Oral Surgery, Wroclaw Medical University, Krakowska 26, 50-425 Wroclaw, Poland; pawel.popecki@umw.edu.pl; 2Department of Oncology, Wroclaw Medical University, Plac Hirszfelda 12, 53-413 Wroclaw, Poland; marcin.zietek@umw.edu.pl; 3Department of Surgical Oncology, Wroclaw Comprehensive Cancer Center, Plac Hirszfelda 12, 53-413 Wroclaw, Poland; 4Department of Maxillofacial Surgery, Medical University of Lodz, 113 S. Zeromski Street, 90-549 Lodz, Poland; marcin.kozakiewicz@umed.lodz.pl

**Keywords:** benign nevus, dysplastic nevus, melanoma, texture analysis, polarized light, non-polarized light, dermoscopy

## Abstract

The differential diagnosis of benign nevi (BN), dysplastic nevi (DN), and melanomas (MM) represents a considerable clinical problem. These lesions are similar in clinical examination but have different prognoses and therapeutic management techniques. A texture analysis (TA) is a mathematical and statistical analysis of pixel patterns of a digital image. This study aims to demonstrate the relationship between the TA of digital images of pigmented lesions under polarized and non-polarized light and their histopathological diagnosis. Ninety pigmented lesions of 76 patients were included in this study. We obtained 166 regions of interest (ROI) images for MM, 166 for DN, and 166 for BN. The pictures were taken under polarized and non-polarized light. Selected image texture features (entropy and difference entropy and long-run emphasis) of ROIs were calculated. Those three equations were used to construct the texture index (TI) and bone index (BI). All of the presented features distinguish melanomas, benign and dysplastic lesions under polarized light very well. In non-polarized images, only the long-run emphasis moment and both indices effectively differentiated nevi from melanomas. TA is an objective method of assessing pigmented lesions and can be used in automatic diagnostic systems.

## 1. Introduction

The differential diagnosis of benign, dysplastic nevi, and melanomas presents a considerable clinical problem. These lesions are similar in a clinical examination but have completely different prognoses and therapeutic management. Benign lesions, such as conventional or common nevi, can be acquired or congenital. Congenital nevi are usually diagnosed at birth, acquired nevi typically appear later, in childhood, and their number increases with age, reaching a maximum around the third or fourth decade of life. In the subsequent decades of life, their number may gradually decrease. Histopathologically, common nevi are formed by melanocytes; they are small and round. Nevus cells can be localized in the epidermis, along the junction of the epidermis and dermis (junctional nevi), present both in the epidermis and dermis (compound nevi) or distributed only within the dermis (intradermal nevi) [1]. Benign nevi do not require any treatment, but large numbers of melanocytic lesions are associated with an increased risk of melanoma with a relative risk of 6.89 in case of 101–120 nevi versus less than 15 [2,3]. Dysplastic nevus (DN), also known as Clark’s nevus or atypical nevus, is clinically characterized by asymmetry, an irregular border, inhomogeneous pigmentation, and a large diameter (more than 5 mm) [4]. In histopathological examination, a dysplastic nevus presents mild to moderate cell atypia, architectural asymmetry, melanocytes of different sizes and shapes that are irregularly disposed within the rete processes [5]. Despite their benign character, dysplastic nevi can clinically mimic malignant lesions. Even in small numbers, their presence indicates an increased risk of melanoma with a relative risk of 6.36 for patients with only five lesions compared to patients without any dysplastic nevi. Their role as precursors of melanoma is also documented, but despite the fact that 25–50% of melanomas develop from benign lesions, the transformation from benign nevi, even dysplastic, to melanoma is very rare. The lifetime risk of nevi evolving into melanoma is 1/3, 164 for men and 1/10,800 for women [3].

Melanoma is one of the most aggressive types of skin cancers that originates in melanocytes. If malignant cells do not extend beyond the epidermis, the tumor is classified as melanoma in situ. Progression of the lesion into the dermis leads to invasive melanoma development. The four most common histologic subtypes of melanoma are superficial spreading melanoma (SSM), accounting for approximately 41–70% of cases, nodular melanoma (NM), occurring in 16–20% of cases, lentigo maligna melanoma (LMM), observed in 2.7–14% of cases, and acral melanoma (AM) occurring in 1–5% of cases. Histologically, SSM presents as atypical melanocytes proliferating in the superficial dermis. NM appears as an exophytic tumor with rapid, vertical growth, that is frequently eroded. LMM presents as a lentiginous proliferation of melanocytes infiltrating the dermis, typically located on the sun damaged areas of the skin. AM appears as a proliferation of spindle melanocytes localized on the subungual or palmoplantar or volar skin [6,7]. Risk factors for melanoma are ultraviolet light exposure, the number of congenital and acquired melanocytic nevi, type of nevi and their size, family history, and genetic susceptibility [8]. Melanoma is relatively rare, accounting for only 4% of all skin cancer cases with a lifetime probability of developing of 3.7% for men and 2.5% for women. Still, at the same time, it is responsible for up to 75–90% of skin cancer deaths [9,10]. The treatment of melanoma is based on the excision of the tumor with surrounding healthy tissue, usually combined with sentinel lymph node biopsy or regional lymph nodes removal. In more advanced cases, surgical procedures can be combined with immunotherapy or targeted therapy with kinase inhibitors as adjuvant systemic therapy [11]. Melanoma assessment and prognosis depend on the depth of infiltration and ulceration in the lesion. The early diagnosis of melanomas with a thickness of below 1 mm results in a 10-year survival of 92% and 50% for melanomas thicker than 4 mm [12]. The statistics show the importance of the early diagnosis of developing melanomas. Although histopathological examination remains the gold standard in identifying melanomas, excision of all pigmented lesions in a patient is usually not possible or may even be excessive and should not be practiced without clear indications. The surgical procedure is always associated with the possibility of certain complications. Every excision may lead to the formation of disfiguring scars, which can be especially important in aesthetic areas, such as the face. Therefore, it is necessary to use diagnostic techniques that provide accurate, early diagnosis and prevent unnecessary excision of benign nevi, misdiagnosed as malignant neoplasms. The assessment of melanocytic nevi usually begins with a visual inspection of the lesion. The examination can be based on the ABCDE rule—evaluation of A—asymmetry, B—border irregularity, C—color heterogeneity, D—diameter of more than 5 mm, and E—evolution of the lesion over time. The simplicity of this technique allows for patient self-evaluation of pigmented nevi. Despite many advantages, the sensitivity of an examination when performed by a dermatologist based on this technique is only 65–70% [13]. Dermoscopy is a non-invasive diagnostic technique that facilitates a better visualization of the lesion surface and subsurface. The assessment of the lesion is carried out with a hand-held optical device with a light source and magnification ranging from 10× to 100×, which improves the sensitivity of the clinical examination up to 70–95% [14]. Although dermoscopy can also be carried out with other, more accurate algorithms such as the pattern analysis, Menzies tool, the seven-point dermoscopy checklist, and the three-point checklist, its effectiveness depends strictly on the dermatologist’s experience. Other available diagnostic techniques for melanoma, such as reflectance confocal microscopy, optical coherence tomography, reflectance spectrophotometry, electrical impedance spectroscopy, and high-frequency ultrasound, offer higher sensitivity and specificity but require specialized equipment and extensive training of the operator so they are able to interpret images properly, making them mostly unavailable [15]. Dermoscopy-based computer-assisted diagnosis (D-CAD) currently offers a sensitivity of 90.1% and specificity of 74.3%. It can accurately identify melanoma in selected populations, but presently D-CAD is unreliable as a stand-alone diagnostic test because of the low specificity [16]. However, as operator-independent tests, a computer-assisted diagnosis system seems to be a highly promising and important direction in research on the diagnosis of pigmented lesions. One of the tools available for D-CAD is texture analysis [17]. Texture analysis (TA) is a mathematical and statistical analysis of pixel patterns of digital images. These patterns can be defined by smoothness, brightness, uniformity, roughness, entropy, granularity, linearity, or randomness. TA includes a wide range of methods and techniques that quantify the interrelationships of pixel, gray-level patterns, and spectral properties of an image. Texture analysis methods can be divided into two main categories—statistical and structural. Statistical methods are based on gray-level statistical analysis that varies over an examined area of the picture. Structural texture techniques focus on the basic elements of the texture’s composition [18]. TA is widely used in medicine to analyze digital images of magnetic resonance, computed tomography, or X-ray [19,20,21].

This study aims to demonstrate the relationship between the texture analysis of digital images of pigmented lesions and their histopathological diagnosis and compare the differences in pictures taken under polarized and non-polarized light. The first null hypothesis we adopted is the lack of differences between benign, dysplastic nevi, and melanomas, both in situ and with invasive techniques. The second null hypothesis is that there is no difference between the texture of images taken under polarized and non-polarized light.

## 2. Materials and Methods

### 2.1. Patients and Lesions

Eighty patients with 94 pigmented lesions were enrolled in this study. All pigmented lesions included in this study were qualified for excision as potentially malignant or atypical by a dermatologist and a surgical oncologist according to the protocol of the Skin Cancer Unit of Wroclaw Comprehensive Cancer Center (M.Z). Before the procedure, all lesions were photographed under polarized and non-polarized light. Then, they were removed under local anesthesia with 1% lidocaine hydrochloride. All lesions were histopathologically examined by experienced histopathologists from the Lower Silesian Cancer Centre specializing in the diagnosis of skin disorders to establish the exact diagnosis and assigned to one of the groups: benign nevi (BN), dysplastic nevi (DN), and melanoma (MM).

The BN group included compound nevi, junctional nevi and intradermal nevi. Other types of benign pigmented lesions were excluded from the study because their characteristic clinical image could have a distorting effect on the results. The DN group included lesions meeting the criteria for dysplastic lesion described in the introduction to the study. The MM group included the superficial spreading melanomas and lentigo maligna melanomas as these types are most similar to benign lesions. In addition, the three-dimensional structure of nodular melanoma usually makes it impossible to take a proper image of its surface, while a different substrate in acral melanoma could interfere with the results of a texture analysis. Pigmented lesions located on the mucosa or the hairy scalp and lesions with erosion or ulceration were excluded from the study. We also excluded lesions whose melanocytic origin was not confirmed in the histopathological examination and lesions larger than 5 cm. Finally, 90 pigmented lesions in 76 patients (40 women and 36 men) were included in the further analysis. 

The mean age of patients was 55, with a median of 55. The youngest patient was 18, and the oldest was 90. The examined lesions included 47 benign nevi, 23 dysplastic nevi, and 20 melanomas (8 in situ and 12 invasive). Fifty-seven lesions were located on the trunk, 15 in the lower limb region, 10 on the upper limb, and eight were located in the head and neck area. The study was approved by the Bioethics Committee of the Wroclaw Medical University, approval number KB—502/2019 (27 May 2019). Informed consent was obtained from all participants. 

### 2.2. Image Preparation

All images were taken with a Canon EOS 77D (Canon Inc., Tokyo, Japan) with a dermoscopy lens DermLite Foto II Pro (3Gen Inc., San Juan Capistrano, CA, USA.) featuring polarized and non-polarized illumination, millimeter-scale, fixed zoom, and focus to ensure stable and reproducible conditions for each photograph. From each photograph, one or more regions of interest (ROI) of 450 × 450 pixels were extracted to represent the most irregular areas of pigmented lesions, collected within their borders, taken from the same site for polarized and non-polarized light images. The number of ROI was determined by lesion area. For the texture analysis, we obtained 166 ROI images for melanomas, 166 for dysplastic nevi, 166 for benign nevi taken under polarized light, and, respectively 166 ROI images for each group in non-polarized light. All ROI images were converted to 8-bit grayscale bitmaps. All graphic operations were done using GIMP version 2.10.30 (GNU Image Manipulation Program—www.gimp.org (accessed on 1 January 2022), free and open-source license).

### 2.3. Texture Analysis

The texture of processed ROI was analyzed in MaZda 4.6 software developed by the University of Technology in Lodz to investigate how the features describe the observed lesions [22]. Data were divided into two main groups, either taken in polarized or non-polarized light, and each main group was subsequently divided into three subgroups with benign nevi (BN), dysplastic nevi (DN) or melanoma (MM) (Figure 1). ROIs from every subgroup were normalized (μ ± 3σ) to share the same average (μ) and standard deviation (σ) of optical density within the ROI. Selected image texture features (entropy and difference entropy from the co-occurrence matrix, and long-run emphasis moment from the run-length matrix) of ROIs were calculated:(1)$$Entropy=-\overset{Ng}{\underset{i=1}{\sum }}\overset{Ng}{\underset{j=1}{\sum }}p\left(i,j\right)log(p\left(i,j\right)$$
(2)$$DifEntr=-\sum_{i=1}^{Ng}p_{x-y}\left(i\right)log\left(p_{x-y}\left(i\right)\right)$$
where Σ—the sum, *Ng*—the number of optical density levels of the image, *i* and *j*—the optical density of pixels 5-pixel distant from one another, *p*—probability, and *log*—standard logarithm [23].
(3)$$LngREmph=\frac{\sum_{i=1}^{Ng}\sum_{k=1}^{Nr}k^{2}p\left(i,k\right)}{\sum_{i=1}^{Ng}\sum_{k=1}^{Nr}p\left(i,k\right)}$$
where Σ—the sum, *Nr*—the number of series of pixels with density level *i* and length *k*, *Ng*—the number of optical density levels of the image, and *p*—probability [24,25].

These three equations were subsequently used to construct the texture index (*TI*) and bone index (*BI*) [26,27]. These indices represented the ratio of the structure diversity measure observed in the image to the measure of uniform longitudinal structures and were calculated as:(4)$$TI=\frac{Entropy}{LngREmph}=\frac{(-\sum_{i=1}^{Ng}\sum_{j=1}^{Ng}p\left(i,j\right)log\left(p\left(i,j\right)\right))\sum_{i=1}^{Ng}\sum_{k=1}^{Nr}p\left(i,k\right)}{\sum_{i=1}^{Ng}\sum_{k=1}^{Nr}k^{2}p\left(i,k\right)}$$
(5)$$BI=\frac{DifEntr}{LngREmph}=\frac{(-\sum_{i=1}^{Ng}p_{x-y}\left(i\right)log\left(p_{x-y}\left(i\right)\right))\sum_{i=1}^{Ng}\sum_{k=1}^{Nr}p\left(i,k\right)}{\sum_{i=1}^{Ng}\sum_{k=1}^{Nr}k^{2}p\left(i,k\right)}$$

The index defined in this way (Equations (4) and (5)) was taken as a measure of structural normality (a measure of healthy insight). 

### 2.4. Statistical Analysis

The Shapiro—Wilk test was used to check normality. Comparisons between different ROIs were performed with a one-way ANOVA or the Kruskal–Wallis test depending on the normality of distribution. A *p* < 0.05 was set to denote statistically significant differences. Statgraphics Centurion version 18.1.12 (StatPoint Technologies Inc., Warrenton, VA, USA) was applied for statistical analyses. We calculated sample size on the basis of power of test. In case of comparison between three groups (for N = 166 in each group) power of test was 98.76%. In the comparison between two groups (N = 166 in each group) we achieved 90.47%. 

## 3. Results

The most common localization of melanomas was in the trunk (9 lesions), then the upper limb (6 lesions), and the lower limb (4 lesions). Only one case of melanoma was observed in the head and neck region. Dysplastic lesions were also most often located within the trunk (17 lesions); other locations were equally frequent (2 lesions for each area). Thirty-nine benign lesions were located on the torso, nine on the lower limb, five in the head and neck region, and only two on the upper limb. The mean age for melanomas (62) was higher than for dysplastic nevi (55) and benign nevi (51). Melanomas were more often found in men (1.5), while benign and dysplastic nevi were more frequent in women (1.5 and 1.3, respectively). 

The data from texture studies of pigmented skin lesions are presented in Table 1 and Figure 1. All analyzed image features and both indicators showed statistically significant differences between the BN, DN and MM groups in polarized light (*p* < 0.001). In non-polarized light only LngREmph and both indices showed statistical differences between the MM and BN groups and between MM and DN groups (*p* < 0.001). However, no statistical significance was found between the DN and BN groups in any of the analyzed parameters in non-polarized light.

The selected feature from the run-length matrix distinguishes the surface features of malignant melanoma of the skin both in polarized and non-polarized light very well. Moreover, (Table 1), in polarized light, the frequency of detection of long series of pixels of similar darkness is higher than in dysplastic nevus (*p* < 0.001), and in this nevus, it is significantly higher than in benign nevus (*p* < 0.001).

The two calculated indices (TI and BI) indicate that the photographic images of malignant melanoma are easily distinguishable from both compared cutaneous nevi. Both TI and BI have statistically significantly lower values (*p* < 0.001) than the image of this malignant neoplasm. This relationship is also detected in polarized light studies (*p* < 0.001). It should also be noted that in dysplastic nevi, TI and BI values were significantly elevated compared to benign nevi (*p* < 0.001).

We also performed an additional analysis within the melanoma subgroup to compare separately invasive and non-invasive (in situ) melanomas. The data from texture studies of in situ and invasive malignant melanoma lesions are presented in Table 2. In situ lesions are significantly different from invasive lesions among all examined features and indices, especially when observed under polarized light.

## 4. Discussion

The mean age of patients with diagnosed melanoma in our study (62) is similar to data available in the literature [28,29,30]. The risk of melanoma increases with age due to the increasing number of spontaneous mutations in melanocytes and the accumulation of UV light exposure, a major risk factor for melanoma development over time. The lower mean age of patients with dysplastic and benign nevi was also observed in other studies and can be explained by a tendency of the acquired nevi number to decrease in time after the third decade of life [31]. Melanoma predisposition in men and both benign and dysplastic nevi in women also find reflection in the literature [31,32]. Although women are more often exposed to UV (ultraviolet) light, e.g., due to the popularity of sunbathing, men are less likely to use UV-protection and less interested in self-observation of their skin, and rarely consult with a dermatologist regarding their pigmented lesions [33,34]. This may be the reason for the higher incidence of melanoma in men and explains the greater number of benign and dysplastic lesions removed in women who seem to be more interested in the health of their skin and more often visit a dermatologist for follow-up.

The location of the lesions was similar in all groups, with a distinct predominance of the trunk, which is probably due to the large surface area of this region. The very low percentage of facial lesions in our study is noteworthy because this area is the most exposed to UV rays. It may be explained by a greater awareness of patients regarding UV protection of this vital aesthetic area and the desire to remove pigmented lesions in a plastic surgery clinic instead of with oncological surgery, to minimize the risk of a disfiguring scar. This last factor also indicates how important it is to improve the clinical diagnosis of melanocytic lesions and prevent unnecessary excisions of benign nevi while maintaining high sensitivity in the diagnosis of melanoma. 

Clinical diagnosis of melanoma using a dermatoscope can increase the sensitivity of the examination from 76 to 92% and specificity from 75 to 95% compared to visual inspection with the naked eye. Still, the accuracy of clinical diagnosis depends strictly on the experience of the dermatologist [15]. To avoid unnecessary surgery and, more importantly, diagnose all cases of melanoma, it is advisable to look for diagnostic methods independent of the operator, giving a measurable, objective, and quantifiable result facilitating an accurate diagnosis. Computer-aided diagnostic systems seem to be the natural direction of diagnostic development. The meta-analysis of the Cochrane Skin Cancer Diagnostic Test Accuracy Group from 2018 established summary sensitivity of current dermoscopy based computer-aided diagnostic systems (Derm-CAD) at 90.1% and summary specificity at 74.3% [16]. They concluded that CAD systems correctly identify melanoma in highly selected populations but are unreliable as the only diagnostic test because of the low and very variable specificity, especially in more diverse populations. This summary shows that there is a need for improvement in this field. Most of the studies for Derm-CAD systems use neural networks and artificial intelligence, comparing the previously entered data with the analyzed data [35,36]. Among other methods, an analysis of texture features, such as contrast, energy, correlation, homogeneity, dissimilarity, inverse difference, and inverse difference moment, can be implemented in Derm-CAD systems [37,38,39,40]. These works are focused mainly on the effectiveness of multifactorial analyses. Still, they do not provide statistical data and the significance of individual elements of the diagnostic system, which may constitute a point of reference for other researchers. 

In this study, we decided to investigate the following three texture features: entropy, difference entropy from the co-occurrence matrix, and the long-run emphasis moment from the run-length matrix and two indicators derived from them: bone index and texture index in terms of the ability to distinguish benign nevi, dysplastic nevi, and melanomas. The texture studies carried out aim to differentiate pathological pigmented skin lesions. The mathematical differentiation from healthy skin is not clinically relevant, as these potentially dangerous pathologies are very visible to the unaided eye. Instead, the clinically important question concerns whether the nevus has undergone dysplastic or atypical activation. An important observation is the convenience of photographed skin lesions under polarized light. Furthermore, each of the presented features or texture indices is statistically different in melanoma of the skin from nevus. Additionally, dysplastic nevi can be distinguished from benign ones. This is important for physicians monitoring the activation of skin nevi [41]. In non-polarized light studies, on the other hand, only some features are helpful (LngREmph, TI, and BI) and only to a limited extent because one can distinguish malignant melanoma from other skin pigmented lesions statistically (but one type of nevus cannot be differentiated from another). These results show unequivocally that the observation of lesions under polarized light facilitates a more accurate and detailed analysis of their structure. It confirms the observations of other authors [42,43]. Decreasing values of entropy and difference entropy with a greater clinical advancement of nevi may result from the progressive expansion of melanocytes in the superficial layers of the skin. The increasing number of pigmented cells in dysplastic nevus and even greater increases in melanoma gradually occupy successive epidermis and dermis layers, blurring the regular pattern typical of benign nevi. This process is also reflected in higher values of long-run emphasis moments. The two proposed indices—bone index and texture index—also emphasize the differences between the studied subgroups, showing statistical differences even within images taken under non-polarized light. These observations are also confirmed by comparing in situ and invasive melanomas. A texture evaluation of images shows clear differences in the appearance of cutaneous malignant melanoma in the in situ phase compared to the invasion phase—invasive melanomas show a significant increase in uniformly black areas. This is evidenced by the high LngREmph values in invasive tumors. It is simultaneously connected with the loss of the chaotic distribution of texture elements monitored by entropy which was measured here in two ways. This indicates a merging of characteristic skin surface features into a uniform solid dark patch. The two proposed indices further emphasize such lesion characteristics. The differences are accentuated much more when observed in polarized light. 

There are no reports in the literature on the use of the examined image parameters in the analysis of dermatoscopic images of pigmented lesions. Only a few studies are available that show the potential of these image features in imaging the pathologic lesions of oral mucosa [19,20]. Zhang et al. used entropy and long-run emphasis in their analysis of CT images to determine prognostic value in early stage non-small cell lung cancer patients receiving stereotactic body radiotherapy [44]. Bonnin at al. used entropy of CT images as predictors of favorable response to anti-PD1 monoclonal antibodies in patients with metastatic skin melanoma [45]. Differential entropy of scanning electron microscope images was used by Hadzik et al. for a comparison of the surface of dental implants [46]. Potentially, the presented parameters can be used to develop a Derm-CAD system for pigmented lesions diagnosis, but it should be stressed that texture analysis is a method sensitive to changes in image exposure. Therefore, for correct results, the photos should always be taken with the same equipment, the same zoom and camera settings, and under the same lighting conditions. Therefore, CAD systems using texture analysis based on retrospectively taken images, due to the unknown origin and unknown conditions under which pictures were taken or analyses of various images of different origin, may lead to inaccurate results when using this technique.

## 5. Conclusions

A texture analysis shows differences between benign, dysplastic nevi, and melanomas.A texture analysis demonstrates differences between in situ and invasive melanomas.Entropy, difference entropy from the co-occurrence matrix and the long-run emphasis moment from the run-length matrix are texture features that can be used for Derm-CAD analysis, and the bone index and texture index can be derived from these three features.Polarized light is superior to non-polarized light in visualizing the details of melanocytic lesions, which leads to a more accurate diagnosis and analysis of the lesion.

## 6. Study Limitations

The limitations of our research concern the sensitivity of the method for the variable conditions of the exposure of the analyzed image, which may affect the results—the compared photographs should always be taken with the same equipment, the same settings, and the same lighting conditions. In addition, the examined method is not well-suited to an evaluation of the ulcerated or damaged lesions, including ones with a large number of hairs, due to a large number of artifacts in the image that may affect the final result.

## Figures and Tables

**Figure 1 jcm-11-02505-f001:**
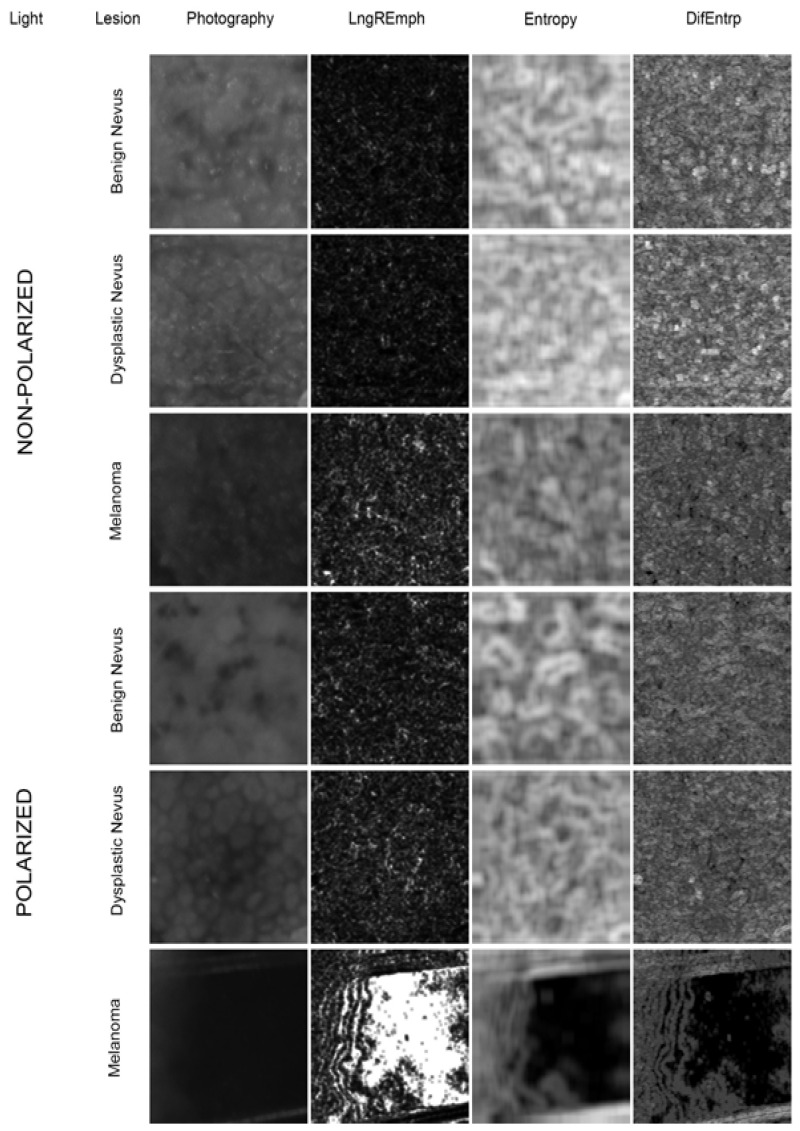
Digital texture analysis of ROI from images of pathological lesions. In the first column from the left are images taken under non-polarized and polarized light of benign pigmented nevus, dysplastic nevus, and melanoma. In the second column from the left, surface intensity distribution maps of the feature describing the frequency of occurrence in the image of long rows of pixels of similar brightness (LngREmph) are presented. In this case, uniformly dark areas were detected. The third and fourth columns show the distribution maps of two texture features describing the randomness of the image structure pattern (Entropy and DifEntrp). The brighter areas define the locations of more chaotically arranged patterns in the skin lesion. Note the complete degradation of the image pattern observed under polarized light in malignant melanomas.

**Table 1 jcm-11-02505-t001:** Results of the investigated five texture features of skin lesions in polarized and non-polarized light.

Imaging	Lesion	LngREmph	Entropy	DifEntrp	TI	BI
Non-polarized Light	BN	2.64 ± 0.80	2.74 ± 0.14	0.98 ± 0.13	1.12 ± 0.29	0.40 ± 0.13
	DN	2.74 ± 0.71	2.77 ± 0.12	0.99 ± 0.12	1.09 ± 0.31	0.39 ± 0.14
	MM	3.16 ± 2.74	2.72 ± 0.15	0.96 ± 0.13	0.97 ± 0.28	0.35 ± 0.12
	Note	*p* = 0.000006	*p* = 0.057	*p* = 0.214	*p* = 0.00004	*p* = 0.0006
		MM > DN			MM < DN	MM < DN
		MM > BN			MM < BN	MM < BN
Polarized Light	BN	5.28 ± 4.32	2.44 ± 0.23	0.76 ± 0.13	0.67 ± 0.33	0.21 ± 0.12
	DN	6.20 ± 3.29	2.38 ± 0.20	0.71 ± 0.10	0.49 ± 0.25	0.15 ± 0.09
	MM	7.84 ± 4.69	2.21 ± 0.29	0.68 ± 0.10	0.39 ± 0.23	0.12 ± 0.07
	Note	*p* = 0.0000001	*p* = 0.0000001	*p* = 0.0000003	*p* = 0.0000001	*p* = 0.0000001
		MM > DN > BN	MM < DN < BN	MM < DN < BN	MM < DN < BN	MM < DN < BN

BN—benign nevus; DN—dysplastic nevus; MM—melanoma.

**Table 2 jcm-11-02505-t002:** Results of the five investigated texture features of in situ and invasive melanoma images taken under non-polarized and polarized light.

Feature	Non-Polarized Light		Statistical Significance	Polarized Light		Statistical Significance
	In Situ	Invasive		In Situ	Invasive	
LngREmph	2.66 ± 0.48	3.65 ± 2.24	*p* = 0.005	6.01 ± 0.10	8.64 ± 4.54	*p* = 0.0000003
Entrory	2.79 ± 0.09	2.67 ± 0.22	*p* = 0.002	2.37 ± 0.27	2.15 ± 0.28	*p* = 0.00000002
DifEntrp	1.00 ± 0.10	0.93 ± 0.16	*p* = 0.001	0.73 ± 0.09	0.67 ± 0.10	*p* = 0.000001
TI	1.09 ± 0.21	0.91 ± 0.17	*p* = 0.004	0.55 ± 0.28	0.32 ± 0.17	*p* = 0.00000008
BI	0.39 ± 0.10	0.32 ± 0.14	*p* = 0.007	0.17 ± 0.09	0.10 ± 0.06	*p* = 0.0000006

TI—texture index; BI—bone index.

## Data Availability

Data are available at pawel.popecki@umw.edu.pl (P.P.) and kamil.jurczyszyn@umw.edu.pl (K.J.).

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
