# Peer review of "Texture Analysis in Diagnosing Skin Pigmented Lesions in Normal and Polarized Light—A Preliminary Report"

_jcm, 2022, doi:10.3390/jcm11092505_

Round 1
Reviewer 1 Report
“Texture analysis in Diagnosing Skin Pigmented Lesions in Normal and Polarized Light – A Preliminary Report” is an exciting contribution that displays the relationship between the texture analysis of images of melanocytic lesions and their histopathological diagnosis. I have the following recommendation:
- The authors state that benign nevi, dysplastic nevi, and melanoma are “very” similar in clinical examination in two places (lines 16, 34). There are similarities clinically; however, there are definitely features to discern between them. The omission of “very” should be considered by the authors.
- Line 59 “extend over the epidermis.” Do they mean “extend beyond the epidermis?”
- Line 62 and 63. Superficial spreading melanoma (low cumulative sun-damage-CSD) and nodular melanoma are mentioned; however, lentigo maligna melanoma (high cumulative sun damage-CSD) and acral melanoma (the other two significant types) are not mentioned. Recommend their mention for completeness.
- Line 131: “All lesions were histopathologically examined….” Details are required. Who examined them? Have they been signed out by fellowship-trained dermatopathologists or general pathologists? Did a dermatopathologist re-examine the cohort to confirm the diagnosis? Alternatively, did the authors just rely on the original reports without engaging the dermatopathologist?
- Line 133: the three categories of benign nevi, dysplastic nevi, and melanoma are acceptable; however, given the diversity within the groups, it is a most straightforward approach. As the authors know, benign nevi can be many types (e.g., compound, intradermal, congenital, blue), and melanomas can have many phenotypes (e.g., superficial spreading, lentigo maligna melanoma, acral, nodular, desmoplastic). These different types will likely have differences in texture analysis. A dedicated paragraph in section 2.1 or a table is required so that the readers can understand the true nature of the cohort.
- The results are well presented in the tables. However, a reader will have to decipher the tables to understand the findings. To engage readers of all levels of training, a simple paragraph that explains the results in more accessible terms is recommended.
- The discussion part is all in one giant paragraph. Breaking it up into smaller paragraphs is recommended.
Author Response
Dear Reviewer
Thank You for Your hard work, that let us to improve our manuscript. We added following corrections:
- The authors state that benign nevi, dysplastic nevi, and melanoma are “very” similar in clinical examination in two places (lines 16, 34). There are similarities clinically; however, there are definitely features to discern between them. The omission of “very” should be considered by the authors.
We deleted “very” in all two indicated places.
- Line 59 “extend over the epidermis.” Do they mean “extend beyond the epidermis?”
We corrected sentence as suggested.
- Line 62 and 63. Superficial spreading melanoma (low cumulative sun-damage-CSD) and nodular melanoma are mentioned; however, lentigo maligna melanoma (high cumulative sun damage-CSD) and acral melanoma (the other two significant types) are not mentioned. Recommend their mention for completeness.
We expanded the paragraph as suggested.
- Line 131: “All lesions were histopathologically examined….” Details are required. Who examined them? Have they been signed out by fellowship-trained dermatopathologists or general pathologists? Did a dermatopathologist re-examine the cohort to confirm the diagnosis? Alternatively, did the authors just rely on the original reports without engaging the dermatopathologist?
We added required details in the paragraph. Health system in Poland does not distinguish subspecialization of dermatopathologist, therefore, we cannot write in the publication that the lesions were examined by a dermatopathologist. However, they were examined by histopathologists of the Wroclaw Comprehensive Cancer Center (Reference oncology center for this region) with very extensive experience in diagnosing skin cancers. Their experience and competences meet all the subspecialization criteria of a dermopathologist distinguished in other countries
- Line 133: the three categories of benign nevi, dysplastic nevi, and melanoma are acceptable; however, given the diversity within the groups, it is a most straightforward approach. As the authors know, benign nevi can be many types (e.g., compound, intradermal, congenital, blue), and melanomas can have many phenotypes (e.g., superficial spreading, lentigo maligna melanoma, acral, nodular, desmoplastic). These different types will likely have differences in texture analysis. A dedicated paragraph in section 2.1 or a table is required so that the readers can understand the true nature of the cohort.
We added more detailed description of individual groups
- The results are well presented in the tables. However, a reader will have to decipher the tables to understand the findings. To engage readers of all levels of training, a simple paragraph that explains the results in more accessible terms is recommended.
We expanded results paragraph as suggested.
- The discussion part is all in one giant paragraph. Breaking it up into smaller paragraphs is recommended.
We divided discussion into smaller paragraphs.
Best regards,
Authors.

Reviewer 2 Report
The following matters are very important for improving the quality and transparency of the paper. • In terms of grammar, the English language of the paper should be improved. • The "Introduction" part of the study should be expanded considering the research objectives, problems, and hypotheses. • The design of the study should be specified in the Materials and Methods section. • If ethics committee approval and informed consent are required for this study, it should be noted within the Materials and Methods section together with the relevant protocol number and date.
• The primary output/endpoint variable(s)/measurements of the study should be defined.• What are the inclusion and exclusion criteria in the study? • Which randomization method was used in the distribution of the individuals included in the study to the groups? • Which blinding (masking) method was used in the study? • Data analysis or Statistical analysis sub-section title should be added to the Materials and Methods. • How was the sample size determined? This information should be explained in the Materials and Methods section. • Which sampling (probable or non-probable, etc.) method was used in the study? • Statistical tests for hypothesis testing and their assumptions should be specified in the statistical analysis of the study in the Materials and Methods section. • The details (version, license number, etc.) of the statistical package(s) or program(s) should be given in the section of "Data Analysis or Statistical Analysis". • It should be explained how the qualitative and quantitative data are summarized under the sub-heading of Statistical Analyzes in the Materials and Methods section of the study. • The exact P values should be added to the table(s) (p=0.25; p=0.03). • Which methods are used to model relationships between variables? • The descriptions and other descriptive values/data should be defined on the tables and shapes. • Are the data subjected to pre-processing? • How were extreme/outlier values in the data determined and resolved? • The number of current references on the subject of the study should be increased. • The discussion section of the research can be expanded by supporting current studies to address the findings of other studies reported with the present findings. • What approaches were used to test the validity of the models? • Which metrics were used in the performance evaluation of the estimates of models/algorithms? • How was the most suitable cut-off point determined using the receiver operator characteristic (ROC) curve analysis? • Which method(s) was/were used to optimize the hyperparameters of models/algorithms?
Author Response
Dear Reviewer
Thank You for Your hard work, that let us to improve our manuscript. We added following corrections:
- In terms of grammar, the English language of the paper should be improved.
The manuscript was checked by certified translator. Certificate of translation was attached.
- The "Introduction" part of the study should be expanded considering the research objectives, problems, and hypotheses.
Study objectives and hypotheses have been already placed in the last paragraph of an introduction in the original manuscript.
- The design of the study should be specified in the Materials and Methods section.
We expanded materials and methods for a more detailed description.
- If ethics committee approval and informed consent are required for this study, it should be noted within the Materials and Methods section together with the relevant protocol number and date.
“The project of the study was approved by the Bioethics Committee of the Wroclaw Medical University, approval number KB—502/2019 (May 27, 2019). Informed consent was obtained from all participants” –these informations has been presented in M&M section (lines 141-144) in original manuscript.
- What are the inclusion and exclusion criteria in the study?
We expanded materials and methods for a more detailed description of the inclusion and exclusion criteria.
- Which randomization method was used in the distribution of the individuals included in the study to the groups?
This study was not randomized - the allocation to each group was based on the result of a histopathological examination.
- Which blinding (masking) method was used in the study?
The study did not require blinding method - all analyzes used in the study were objective and based on quantifiable parameters and did not depend in any way on the patient or researcher. All cases were diagnosed in the same way.
- Data analysis or Statistical analysis sub-section title should be added to the Materials and Methods.
All informations about statistical analysis were already placed in original manuscript, in M&M subsection: Statistical Analysis.
- How was the sample size determined? This information should be explained in the Materials and Methods section.
We calculated sample size on the base of power of test. In case of three groups comparison (for N=166 in each group) power of test was 98.76%. In case of comparison of two groups (N=166 in each group) we achieved 90.47%. We added this information to the M&M section.
- Which sampling (probable or non-probable, etc.) method was used in the study?
In this study we used purposive sampling (non-probable) - patients with suspicious pigmented lesions eligible for surgical excision
- Statistical tests for hypothesis testing and their assumptions should be specified in the statistical analysis of the study in the Materials and Methods section.
The Shapiro–Wilk test was used to check normality. Hypothesis Zero: no difference between average value among groups (ANOVA), or in cases there no normal distribution were detected, the Kruskal-Wallis test and Hypothesis Zero: no difference in median value among groups.
- The details (version, license number, etc.) of the statistical package(s) or program(s) should be given in the section of "Data Analysis or Statistical Analysis".
All of these informations were already presented in the original manuscript: Statgraphics Centurion version 18.1.12 (StatPoint Technologies Inc., Warrenton, VA, USA)
- The exact P values should be added to the table(s) (p=0.25; p=0.03).
We added p value in the tables.
- Which methods are used to model relationships between variables?
Simple regression method
- The descriptions and other descriptive values/data should be defined on the tables and shapes.
All of these informations are already presented in the original manuscript.
- Are the data subjected to pre-processing?
After image acquisition, there were no data pre-processing.
- The number of current references on the subject of the study should be increased.
We increased number of references.
- The discussion section of the research can be expanded by supporting current studies to address the findings of other studies reported with the present findings.
We add references to other studies, but there are only limited studies using similar techniques and parameters and they are concerning other fields.
- What approaches were used to test the validity of the models?
We didn't produce any models in this study. The study is focused on showing the relationship between the examined image features within individual groups
- Which metrics were used in the performance evaluation of the estimates of models/algorithms?
We didn't produce any models and algorithms in this study. The study is focused on showing the relationship between the examined image features within individual groups
- How was the most suitable cut-off point determined using the receiver operator characteristic (ROC) curve analysis?
We did not perform ROC curve analysis in this study.
- Which method(s) was/were used to optimize the hyperparameters of models/algorithms?
We didn't produce any models and algorithms (in machine learning) in this study. The study is focused on showing the relationship between the examined image features within individual groups.
Best regards,
Authors.

Round 2
Reviewer 1 Report
The concerns raised in the first round of review are addressed by the authors. Thank you.
Reviewer 2 Report
Acceptable
This manuscript is a resubmission of an earlier submission. The following is a list of the peer review reports and author responses from that submission.